



**Long‒range transport of anthropogenic air pollutants into the**
**marine air: Insight into fine particle transport and chloride depletion**
**on sea salts**
Liang Xu[1], Xiaohuan Liu[2], Huiwang Gao[2], Xiaohong Yao[2], Daizhou Zhang[3], Lei Bi[1], Lei Liu[1],
Jian Zhang[1], Yinxiao Zhang[1], Yuanyuan Wang[1], Qi Yuan[2], Weijun Li[1,*]
[1]Department of Atmospheric Sciences, School of Earth Sciences, Zhejiang University, Hangzhou
310027, China
[2]Key Laboratory of Marine Environment and Ecology, Ministry of Education, Ocean University of
China, Qingdao 266100, China
[3]Faculty of Environmental and Symbiotic Sciences, Prefectural University of Kumamoto,
Kumamoto 862-8502, Japan
*Corresponding author: W. Li (liweijun@zju.edu.cn)
**Abstract**
Long-range transport of anthropogenic air pollutants from East Asia can affect the
downwind marine air quality during spring and winter. Long-range transport of
continental air pollutants and their interaction with sea salt aerosols (SSA)
significantly modify the radiative forcing of marine aerosols and influence ocean
biogeochemical cycling. Previous studies poorly characterize variations of aerosol
particles along with air mass transport from the continental edge to the remote ocean.
Here, the research ship R/V Dongfanghong 2 traveled from the eastern China seas



(ECS) to the northwestern Pacific Ocean (NWPO) to understand what and how air
pollutants were transported from the highly polluted continental air to clean marine air
in spring. A transmission electron microscope (TEM) was used to find the long-range
transported anthropogenic particles and the possible Cl-depletion phenomenon of SSA
in marine air. Primary and secondary anthropogenic aerosols were identified and
dramatically declined from 87% to 8% by number from the ECS to remote NWPO.
For the SSA aging, 86% of SSA particles in the ECS were identified as fully aged,
while the proportion of fully aged SSA particles in the NWPO decreased to 31%. The
result highlights that anthropogenic acidic gases in the troposphere (e.g., $SO_2$, $NO_x$,
and volatile organic compounds) were transported longer distances compared to the
anthropogenic aerosol and could exert a significant impact on marine aerosols in the
NWPO. These results show that anthropogenic particles and gases from East Asia
significantly perturb aerosol chemistry in marine air. The optical properties and cloud
condensation nucleation of the modified SSA particles should be incorporated into the
more accurately modeling of clouds in the ECS and NWPO in spring and winter.




## 1. Introduction


Marine aerosols play an important role in the global aerosol emission budget and
greatly impact the Earth's radiative forcing and biogeochemical cycling (O'Dowd
Colin and de Leeuw, 2007). Sea salt aerosol (SSA) is one crucial component of
marine aerosols, especially in the remote marine atmosphere (Lewis and Schwartz,
2004). SSA is composed of Na and Cl with minor amounts of Mg, Ca, K, and S (Li et
al., 2016a). Fresh SSA usually has the form of a cubic NaCl core associated with
$MgCl_2$ and $CaSO_4$ coating (Chi et al., 2015). SSA in polluted air can serve as reactive
surfaces through the uptake of acidic gaseous $SO_2$, $NO_x$, and organic acids, releasing
gaseous reactive chlorine compounds (chloride depletion) (Laskin et al., 2012; Yao
and Zhang, 2012). The chloride depletion further releases Cl- and I-containing
compounds into the air which could drive marine new particle formation (He et al.,
2021; Yu et al., 2019). Moreover, the SSA aging processes can transform fresh SSA
into partially aged SSA and finally into fully aged SSA that mainly contain $NaNO_3$,
$Na_2SO_4$, and organic sodium salts (Chi et al., 2015; Laskin et al., 2012). These
products from the chemical aging processes also modify hygroscopic properties of
individual SSA (Cravigan et al., 2020; Ghorai et al., 2014). Some studies found that
the aged SSA could alter global climate directly by scattering incoming solar radiation
or indirectly by acting as cloud condensation nuclei (CCN) or ice nuclei (IN) in
marine air (Hu et al., 2005; Murphy et al., 1998; Pierce and Adams, 2006).
It is well known that the interaction of the continental-marine air masses not only
releases some active substances into marine air but also supplies many nutrients (e.g.,



Fe, N, and P) for biological growth (e.g., plankton) to the ocean's surface (Li et al.,
2017; Shi et al., 2012). Some studies have reported that the continental anthropogenic
and natural pollutants can be carried to remote marine air through long-range
atmospheric transport (Guo et al., 2014; Li et al., 2017; Moffet et al., 2012; Uematsu
et al., 2010). The continental pollutants that are deposited into the ocean increase the
nutrient input to the seawater, and finally alter the primary productivity in the open
sea (Fu et al., 2018; Luo et al., 2016; Mahowald et al., 2018; Shi et al., 2012).
Moreover, previous studies found that large amounts of light-absorbing aerosols (e.g.,
black carbon and brown carbon) from continental polluted air can be transported into
the open ocean air and significantly influence the radiative balance of the marine
boundary layer (Kang et al., 2018; Kondo et al., 2016; Ueda et al., 2018; Zhang et al.,
2014). Therefore, it is significant to understand the physicochemical properties of
continental anthropogenic aerosol particles in marine air.
The eastern China seas (ECS: the Yellow Sea and the East China Sea) and the
northwestern Pacific Ocean (NWPO) can be affected by the Asian continental air
masses under the prevailing westerly winds in winter and spring (Uematsu et al., 2010;
Uno et al., 2009). At present, there have been many in-depth studies on the
physicochemical properties of aerosols in air masses before they leave the Asian
continent. For example, Li et al. (2014) collected aerosol particles at a background
site in the Yellow River Delta and determined their physicochemical properties
before leaving the Asian continent. Feng et al. (2012) studied the sources and
formation pathways of $PM_{2.5}$ at Changdao Island, a resort island in Bohai Sea/Yellow



Sea, which is in the transport path of the Asian continental outflow to the Pacific
Ocean. Shi et al. (2019) investigated aerosol particles from Asian continental outflow
in Qingdao and found the solubility of phosphorus was related to the sources and
atmospheric acidification processes. However, these atmospheric field observations
were limited to some isolated continental sites.

Shipboard observations are an effective way to study marine aerosol properties in

remote areas. Using the single particle analysis method (i.e., electron microscope),
previous shipboard atmospheric studies have observed chloride depletion and sulfur
enrichment in SSA particles from the marine boundary layer (Bondy et al., 2017;
McInnes et al., 1994; Mouri and Okada, 1993). At a coastal city in southwestern Japan,
Zhang et al. (2003) found that dust particles from Asian continent could mix with SSA
particles in the marine atmosphere and further restrained chloride depletion from the
sea-salt component in the particles. However, these studies did not examine how
anthropogenic pollutants influence the physicochemical properties of SSA from the
margin sea air to the remote marine air. Furthermore, they cannot continuously trace
the changes of anthropogenic aerosol particles along the pollutants' transport path to
the remote NWPO. These information is critical to comprehensively understand the
influence of continental anthropogenic air pollutants on the marine air.

To achieve this aim, the research ship R/V Dongfanghong 2 was designed to

cruise from the ECS to the NWPO so that we could understand what and how air
pollutants are transported from the highly polluted continental air to clean marine air
in spring. After the cruise, a transmission electron microscope was used to obtain





composition, size, morphology, and mixing states of marine aerosol particles. Based
on this information, we compared aerosol characteristics over the ECS and the remote
NWPO. Furthermore, we also discussed how the continental air masses influence
marine aerosols in the ECS and the NWPO air.

**2. Experiments**
**2.1 Aerosol sampling and analyses**
Aerosol samples were collected on board the R/V Dongfanghong 2 during the
cruise from 17 March to 22 April 2014. The cruise path crossed the ECS and the
NWPO (Figure 1). Aerosol particles were sampled onto copper TEM grids (carbon
type-B, 300-mesh copper; Tianld Co., China) using a DKL-2 sampler (Genstar
Electronic Technology, China). The sampler was equipped with a single-stage
impactor with a 0.5 mm diameter jet nozzle at an airflow of 1.0 L/min. If the particle
density is 2 g cm$^{-3}$, the collection efficiency of the sampler is 50% for particles with a
260 nm aerodynamic diameter. All the samples were collected at the ship's bow to
avoid contamination from the exhaust. During the same cruise, the short-period
contribution from contamination of particulate matter was less than 3% (Luo et al.,
2016). The sampling duration varied from 2 to 3 min to avoid individual particles
overlap on the substrate. After the collection, all the samples were sealed in dry plastic
capsules and stored in a desiccator at 25 ℃ and 20 ± 3% relative humidity (RH) for
further analysis.
The aerosol particles were analyzed by a JEOL JEM-2100 transmission electron



microscope (TEM) operated at 200 kV. The chemical elements (heavier than carbon,
$Z \geq 6$) were semi-quantitatively detected by an energy-dispersive X-ray spectrometer
(EDS) (Oxford Instruments, UK). The iTEM software (Olympus Soft Imaging
Solutions GmbH, Germany) was used to analyze individual particles in the TEM
images and obtain their projected area, perimeter, aspect ratio, and equivalent circle
diameter (ECD).

A total of 22 samples were analyzed in this study. The location of each sample is

shown in Figure 1. The details about sampling dates, times, and meteorological
conditions for each sample are listed in Table S1. Due to the influence of the
westerlies, the ECS is frequently affected by the air pollutants' transport from Asia in
spring (Shi et al., 2019). The NWPO is less affected by the transport of continental
pollutants because of the remote distance (Zhang et al., 2018). According to the
sample locations, we defined two sample categories along with the cruise path (Figure
1): 11 samples in the ECS and 11 samples in the NWPO.
**2.2 Air mass backward trajectories**

The NOAA HYSPLIT (Hybrid Single Particle Lagrangian Integrated Trajectory)

trajectory model (Stein et al., 2015) was applied to calculate the backward trajectories
for the investigation of air mass sources and transport paths. The total run time was
set at 48 hours. We selected an altitude of 500 m as the endpoint in each backward
trajectory. In this study, we obtained 21 backward trajectories.

**3. Results**



### 3.1 Classification and relative abundance of aerosol particles


The TEM analyses provided the morphology, mixing state, and composition of
individual particles. In this study, we analyzed 3,734 particles in 22 samples (2,770
particles collected in the ECS and 964 particles in the NWPO). Aerosol particles were
classified into seven types based on their composition and morphology: sulfur-rich
(S-rich), organic matter (OM), soot, metal, fly ash, mineral, and sea salt. S-rich
particles are considered as secondary inorganic particles (e.g., $(NH_4)_2SO_4$ and
$NH_4NO_3$), which are formed from their gaseous precursors, such as $SO_2$, $NO_x$, and
$NH_3$. OM particles mainly contain C and certain O and Si. Here we observed two
kinds of OM particles: spherical or irregular primary organic matter (POM) particles
and secondary organic matter (SOM) particles. The POM is directly emitted from the
combustion of fossil fuel and biomass and SOM is formed from volatile organic
compounds (VOCs) or the oxidized POM in the atmosphere (Li et al., 2016a; Wang et
al., 2021). The SOM is normally mixed with S-rich particles (Li et al., 2016b). Soot
particles (i.e., black carbon) are chain-like aggregates of carbonaceous spheres,
mainly containing C and minor O. Metal particles mainly contain Fe, Zn, and Pb, and
fly ash particles contain Si, Al, and minor Ca and Fe. Metal and fly ash particles both
display spherical shapes and are directly emitted from heavy industrial activities such
as power plants and steel factories (Li et al., 2017). Mineral particles are composed of
Si, Al, Ca, and Fe and present irregular shapes (Figure 2a). Mineral particles originate
from arid deserts (e.g., Sahara and Gobi), roads, and construction activities in the
continental areas. Sea salt aerosol (SSA, Figure 2b) is from the bursting of air bubbles





resulting from the waves breaking. SSA is mainly composed of Na and Cl, with minor
Mg, Ca, K, and S.
The high-resolution TEM could see through the thin materials on the substrate, so
the inner mixing structure of different aerosol components in individual particles can
be directly identified (Li et al., 2016b; Riemer et al., 2019). We found that most of the
individual non-SSA particles collected in marine air contained two or more different
types of anthropogenic aerosols. To elucidate the mixing structure of the non-SSA
particles, we further defined six types of non-SSA particles: S-metal (Figure 2c), S-fly
ash (Figure 2d), S-soot (Figure 2e), OM coating (Figure 2f), OM-S (Figure 2g), and
S-rich (Figure 2h). In the ECS, anthropogenic aerosols accounted for 87% of all
particles by number fraction, including S-rich for 42 %, S-soot for 21%, S-metal/fly
ash for 8%, OM-S for 6%, and OM coating for 10%. Only 8% of the observed
particles in the ECS were SSA particles. The remaining 5% was identified as mineral
particles. Interestingly, SSA particles became the dominant aerosol at 90% in the
NWPO and anthropogenic aerosols only accounted for 8%, suggesting that marine
emissions became the primary aerosol source in the NWPO. Therefore, there are large
differences between aerosol particles in the ECS and the NWPO (Figure 3).

**3.2 Variations of aerosol particles from the ECS to the NWPO**
Figure 3 shows variations of aerosol particles along with the cruise pathway from
the ECS to the NWPO. Mineral particles can only be transported from continental
areas. Figure 3 shows that higher number fractions of mineral particles always





occurred when the sampling sites were close to eastern China. The number fraction of
mineral particles rose to 15% in the coastal air during the cruise (Figure 3). The
proportion of mineral particles decreased from 15% to 3% for the samples collected in
the ECS. When the ship traveled eastward into the NWPO, the proportion of mineral
particles dropped to a low level (2%). In contrast, the proportion of mineral particles
increased again when the ship returned to the ECS. Altogether, the number fraction of
mineral particles was 5% in the ECS, twice as high as that in the NWPO, suggesting
that the ECS and NWPO were influenced by westerlies during the sampling period.
Indeed, Figure 1 shows that most of 48h backward trajectories of air masses in the
ECS were sourced from eastern China and that some of the 48h back trajectories of air
masses in the NWPO crossed Japan.

Figure 3 shows that S-metal and S-fly ash (two typical anthropogenic aerosol

particles) displayed variation similar to the mineral particles. Number fractions of
S-metal and S-fly ash particles in the ECS samples ranged from 17% to 2% with the
average value at 8%, but only 0.3% for S-metal and S-fly ash particle was found in
the NWPO. In a word, we conclude that aerosol particles from the Asian continent
directly exert much greater impacts on the ECS than the NWPO.
**3.3 Comparison of anthropogenic secondary aerosol particles**

Secondary S-rich and OM coating particles are normally considered as arising

from the conversion of anthropogenic gaseous pollutants (e.g., $SO_2$, $NO_x$, $NH_3$, and
VOCs) (Li et al., 2021). TEM observations clearly identified secondary particles and
showed their variation of number fraction in the samples (Figure 3). Our results show



that number fractions of S-rich particles were dominant in the ECS samples with the
range of 32%-71%, but their fractions decreased to 5% in the NWPO. The results
indicate that secondary particles in the ECS were strongly influenced by
anthropogenic pollutants transported from eastern China. Furthermore, we noticed
that secondary aerosol particles were frequently mixed with some typical fine primary
anthropogenic particles (e.g., soot, fly ash, and metal) and formed S-soot/S-fly
ash/S-metal particles (Figure 4). As a result, we conclude that the anthropogenic gases
or aerosol pollutants in the continental air masses significantly influence the
downwind air quality of the ECS, but they have a minor impact on the NWPO air.
It should be noted that OM coating particles were frequently found in the ECS but
barely observed in the NWPO. In other words, S-rich particles in the NWPO had no
typical OM coating, although S-rich particles accounted for ~5% in the NWPO
samples. Figure 4b shows that these S-rich particles in the NWPO only had one
dominant size range smaller than 1 μm, which is different from the larger and broader
size distribution of S-rich particles in the ECS. In addition, we noticed the particularly
high fraction of S-rich particles in Sample #11 and #12 collected in the NWPO (15%
and 24%). These results indicate that these S-rich particles likely formed in the
NWPO air. Coincidently, Zhu et al. (2019) observed a new particle formation event in
the same area and proposed that the event was likely caused by long-range transported
continental gases (e.g, $SO_2$, $NO_x$, and VOCs).
**3.4 Aging of sea salt aerosols**
As the dominant aerosol particles in marine air, SSA particles accounted for



70%-98% in the NWPO. SSA could serve as reactive surfaces for heterogeneous and
multiphase chemical reactions in the atmosphere, and these reactions also could alter
the morphology and composition of SSA (Athanasopoulou et al., 2008; Chi et al.,
2015; Laskin et al., 2012). Based on the morphology and composition of SSA, we
further classified SSA into three categories: fresh SSA, partially aged SSA, and fully
aged SSA (Figure 5).

The fresh SSA did not experience any chemical modification in the atmosphere.

TEM images of fresh SSA indicate the cubic NaCl core and coating composed of
$MgCl_2$ and $CaSO_4$ (Figure 5a). The NaCl core only contains Na and Cl, with the
atomic ratio of Na to Cl close to 1:1 (Figure 5d). The major components in the coating
are Mg, O, S, Cl, and Ca (Figure 5e), thus, we infer their molecular forms as $MgCl_2$
and $CaSO_4$ (Buseck and Pósfai, 1999; Chi et al., 2015; Geng et al., 2010; Pósfai et al.,

1994).

The partially aged SSA represent those SSA particles that undergo chemical

modification but still retain part of the NaCl core (Figure 5b). The morphological
differences can be observed between the fresh SSA and partially aged SSA. The NaCl
core still persists in the partially aged SSA but cannot keep its regular cubic shape.
Meanwhile, the coating composition turns into O, Na, Mg, Ca, and S, with decreasing
Cl (Figure 5f). The SSA aging is attributed to the Cl-depletion phenomena, which can
be expressed as follows (Laskin et al., 2012):

$NaCl\ (aq) + HA\ (aq,\ g) \rightarrow NaA\ (aq) + HCl\ (aq,\ g)$

where HA represent atmospheric acids (e.g., $H_2SO_4$, $HNO_3$, and methanesulfonic



acid). NaCl in the SSA could react with inorganic (e.g., $HNO_3$ and $H_2SO_4$) or organic
acid (e.g., methanesulfonic acid), releasing volatile HCl (g) to the atmosphere, leading
to depletion in chloride and enrichment in corresponding sodium salts.
We define the fully aged SSA as particles whose NaCl cores have been
completely transformed into $NaNO_3$ and $Na_2SO_4$. Figure 5c shows that the NaCl cores
in the fully aged SSA entirely disappeared, leaving a rounder shape. The Cl element
was no longer detected in the fully aged SSA and the major aerosol components were
the mixture of $NaNO_3$ and $Na_2SO_4$ (Figure 5g).
To evaluate composition differences of SSA, we present triangular diagrams of
Na-Cl-S weight ratio based on the EDS. Figure 6a shows that the fresh SSA is around
the NaCl, the partially aged SSA is distributed in the center of triangular, and the fully
aged SSA is around $NaNO_3$ and $Na_2SO_4$. Figure 6b shows that there were large
variations of the SSA components in the ECS and NWPO. Our results show that most
of the aged SSA in the ECS were the mixture of $NaNO_3$ and $Na_2SO_4$, suggesting that
SSA in the ECS underwent heterogeneous reactions and become fully aged SSA. SSA
in the NWPO is widely distributed between NaCl and $NaNO_3/Na_2SO_4$ in the
triangular diagram, suggesting that these were partially aged SSA particles.
Figure 7 shows the relative abundance of SSA at different sampling sites from the
ECS to the NWPO. In the ECS, 86% of SSA particles were fully aged. As the ship
traveled into the NWPO, fully aged SSA particles decreased to 31%, meanwhile the
proportion of fresh SSA increased to 55%. The size distribution of SSA particles
shows that the proportion of fresh SSA increased with the increase of particle size



from 0.05 to 5 μm (Figure 8). On the contrary, the proportion of fully aged SSA
decreased with the increase of particle size from 0.05 to 5 μm. Overall, partially and
fully aged SSA accounted for 61% of SSA particles smaller than 3 μm, while fresh
SSA dominated (81%) in SSA particles larger than 3 μm. Figure 8 also reveals that 94%
of the fully aged SSA particles were smaller than 3 μm.

**4. Discussion**

Abundant BC, metal, and fly ash particles in the ECS show that long-range

transport of anthropogenic aerosol particles from the polluted continental areas
constantly influence the ECS air during the spring and winter. Moreover, the existence
of OM coating particles (10%, Figure 3) in the ECS indicates that secondary
sulfate/nitrate particles underwent aging process and formed the OM coating during
their transport (Li et al., 2021). This result suggests that the long-range transported air
masses from continental areas brought abundant anthropogenic gases such as $NO_x$,
$SO_2$, and VOCs into marine air. Indeed, the HYSPLIT backward trajectories show
that the air masses in the ECS were mainly from eastern China (Figure 1). Under the
influence of the westerlies, a large number of anthropogenic and natural pollutants are
transported to the ECS and further influence its air quality.

Figure 3 shows that both primary and secondary anthropogenic aerosols were

relatively low in the NWPO (8%), suggesting a slight influence from continental
aerosols. However, we observed severe Cl-depletion in SSA in some samples in the
NWPO. The Cl-depletion of SSA is mainly caused by the heterogeneous reactions



with acidic gaseous pollutants in marine air (Chi et al., 2015; Hsu et al., 2007; Laskin
et al., 2012). By comparing the aging degree of SSA and air mass backward
trajectories, we found that the air masses with a relatively high proportion of fully
aged SSA particles (#13 and #15) mostly formed due to the anthropogenic gaseous
pollutants (e.g., $SO_2$ and $NO_x$) from northwest China and Japan (Figure S1). The
result suggests that aerosol particles and gases might have different transport distances.
Aerosol particles could be significantly removed by dry or wet deposition. However,
the anthropogenic gases can be transported further to the NWPO air (Figure 9). Over
the western Pacific, Koike et al. (2003) also found that anthropogenic gaseous
pollutants (e.g., $NO_x$ and $SO_2$) from the East Asia have higher transport efficiency
than aerosols from the same region. Meanwhile, the surface changes of SSA can be
considered as a potential indicator for anthropogenic gaseous pollutants in remote
marine air.
Figure 6 shows a higher percentage of Cl-free SSA particles (on Cl = 0 line) in
the ECS than those in NWPO, suggesting modifications of SSA in the ECS were
much more severe than them in the NWPO. Referred to Zhang et al. (2003), we
further provide the dash line that Na:Cl:S was changed only by reaction with $H_2SO_4$.
Thus, particles above the dash line represent that Cl in these SSA particles was
replaced by S deposition and other chemical processes (e.g., $HNO_3$ and organic acids).
The number of fully aged SSA particles above the dash line was further counted.
Surprisingly, we found that Cl-depletion in 70% of the fully aged SSA particles in the
ECS was not only caused by S deposition, but the fraction increased to 87% in the





NWPO (Figure 6). The result indicates that a higher proportion of the fully aged SSA
particles in the NWPO were reacted with diverse acids besides $H_2SO_4$.

It is well known that organic acids could also be a reason of Cl-depletion in SSA

(Chi et al., 2015; Laskin et al., 2012). It is well known that the oxidation of dimethyl
sulfide (DMS) emitted from phytoplankton contributes to the Cl-depletion in SSA
(Sievering et al., 2004). Both anthropogenic acid gases and DMS in the ECS were
found to be higher than that in the NWPO (Figures S1 and S2). Compared with the
ECS, we found that DMS in our NWPO shipping area had lower mass concentrations
and minor fluctuations (3-6 $\times10^{-11}$ kg $m^3$, Fig S2). However, the percentages of aged
SSA differed widely (from 23% to 91%) within the NWPO. We deduce that the low
concentration of the DMS only slightly modified part of SSA in the NWPO, but it was
not enough to influence all the SSA. Moreover, Zhu et al. (2019) reported that
secondary sulfate particles in the NWPO were mostly came from the long-range
transported acidic gases during the sampling period. The contribution of DMS to
non-sea-salt sulfate is less than 6% in the remote ocean of the northern hemisphere
(George et al., 2008; Quinn et al., 1990; Savoie et al., 1994). Therefore, a large
proportion of aged SSA particles in the NWPO should be mainly attributed to the
anthropogenic acidic gases from long-range transport.

Previous studies found that SSA emissions increased with the increase of wind

speed (Feng et al., 2017; Pant et al., 2008; Shinozuka et al., 2004). In our study, the
proportion of fresh SSA increases with increasing wind speed (Figure 7), consistent
with the aforementioned studies. SSA particles of a smaller size have lower dry



deposition velocities and longer lifetimes in the air (Lewis and Schwartz, 2004),
which could enhance the Cl-depletion. This could be the reason that partially and fully
aged SSA was mostly in the smaller size range (<3 μm) (Figure 8). On the contrary,
the newly emitted coarse SSA particles with high dry deposition velocity are more
likely to deposit to the ocean, resulting in less reacted SSA. As a result, the fresh SSA
from local sea spray was mostly found in the coarse size range (larger than 3 μm) in
our samples (Figure 8). This result suggests that it is crucial to study aging process
among the size-resolved SSA, especially particles smaller than 3 μm.

In the future, we need to pay more attention to the influence of anthropogenic

gaseous pollutants on the SSA aging in remote marine air. On the one hand, the aging
processes could modify the hygroscopicity of SSA, determining their morphology and
phase state in the humidified marine environment, in the end directly affecting optical
properties of SSA (Wang et al., 2019). On the other hand, the hygroscopicity change
due to SSA aging could alter CCN activity, and indirectly affect global climate (Hu et
al., 2005; Murphy et al., 1998; Pierce and Adams, 2006). The SSA aging processes
could release more Cl- and I-containing compounds into air, which might lead to
marine new particle formation and growth (He et al., 2021). Meanwhile, the SSA also
serves as an important sink for the anthropogenic acidic gases in remote marine areas
(Chi et al., 2015; Laskin et al., 2012). Thus, in future research, it would be crucial to
quantify the anthropogenic acidic gases scavenged by SSA.

**5. Conclusions**



Individual aerosol particles were collected from 17 March to 22 April, 2014 on
board of the ship R/V Dongfanghong 2 from the ECS to the NWPO. We classified
aerosol particles based on their composition, morphology, and mixing state: mineral,
sea salt, S-metal, S-fly ash, S-soot, OM coating, OM-S, and S-rich. Microscopic
analysis showed that anthropogenic aerosols accounted for 87% of the total particle
number in the ECS. In particular, higher proportions of secondary particles (i.e.,
S-rich particles, 42%, and OM coating particles, 10%) were found in the ECS.
Meanwhile, primary and secondary anthropogenic aerosols are relatively low in the
NWPO (8%).
TEM observations revealed that SSA particles were the most abundant in the
NWPO atmosphere, accounting for 90% of all analyzed aerosol particles. The
Cl-depletion of sea salt aerosol (SSA) particles caused by the heterogeneous reactions
with acidic gaseous pollutants was further observed. Three types of SSA particles,
fresh, partially aged, and fully aged were classified. Fully aged SSA particles were the
dominant SSA in the ECS (86%), while fully aged SSA particles decreased to 31% in
the NWPO. The severe aging of SSA (partially and fully aged, at most 88% of SSA)
was still found in the NWPO, despite there being only minor anthropogenic aerosol
particles. These results show that aerosol particles from the continent air might be
removed by dry and wet deposition, but the air pollutants were transported further to
the NWPO. The aging of SSA particles has important effects on their hygroscopic and
optical properties, one effect being the promotion of heterogeneous reaction with
acidic gases in the NWPO. Our observations show that more attention should be given



to the influence of anthropogenic gaseous pollutants on the Cl-depletion on SSA in
remote marine areas.

**Supplement**


The supplement related to this article is available online at:

**Author contributions**


LX and WL conceived the study and wrote the article. The sampling during the
research cruise was organized by XL, HG, and XY. LX, WL, LL, JZ, YZ, YW, and
QY carried out TEM analyses of individual particles. DZ and LB contributed to the
improvement of this paper. All authors reviewed and approved the paper.

**Competing interests**


The authors declare that they have no conflict of interest.

**Acknowledgements**


We thank Peter Hyde for his editorial comments. We acknowledge the NOAA Air
Resources Laboratory for the provision of the HYSPLIT transport and dispersion
model and READY website (http://www.ready.noaa.gov) used in this publication.

**Financial support**


This research was supported by the National Natural Science Foundation of China



(grant nos. 42075096, 91844301, and 41807305), the National Key R&D Program of
China (grant no. 2017YFC0212700), Zhejiang Provincial Natural Science Foundation
of China (grant no. LZ19D050001), and China Postdoctoral Science Foundation
(grant no. 2019M662021).

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



**Figures**

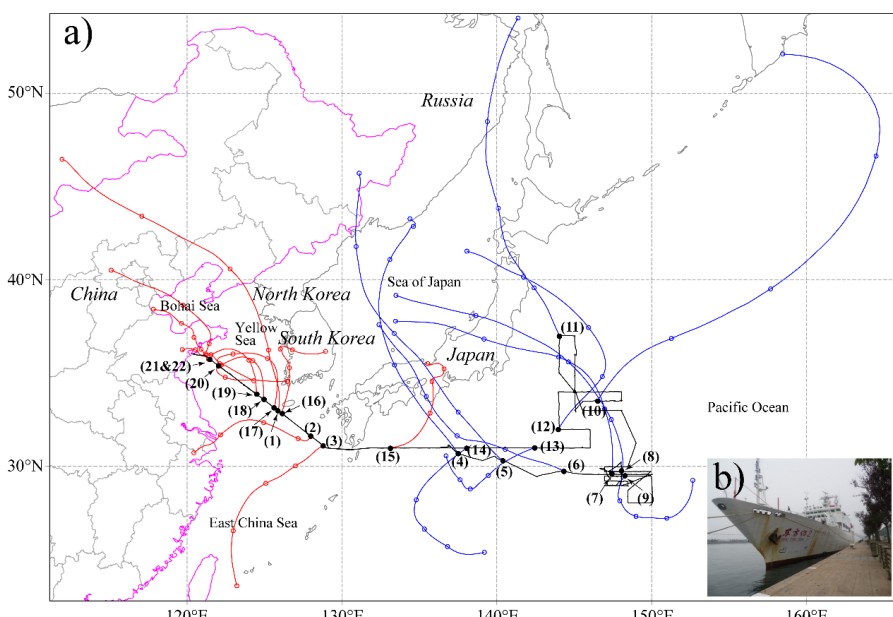

Figure 1. (a) Map of the cruise track (black line) and 48 h air mass backward
trajectories (red and blue lines) arriving at 500 m above ground level at sampling
locations. The interval between two circle symbols is 6 h. The number represents the
sample ID in Table S1. (b) Photo of the R/V Dongfanghong 2.

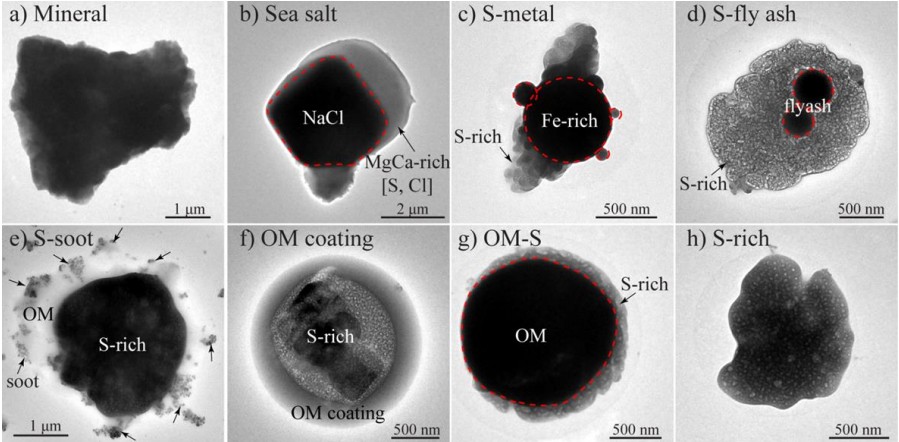






Figure 2. Transmission electron microscope (TEM) images of different types of
aerosol particles: (a) mineral; (b) sea salt; (c) metal particles mixed with sulfate; (d)
fly ash particles mixed with sulfate; (e) soot particles mixed with sulfate; (f)
secondary organic matter (OM) coating on sulfate; (g) primary OM particle mixed
with sulfate; (h) S-rich particle.

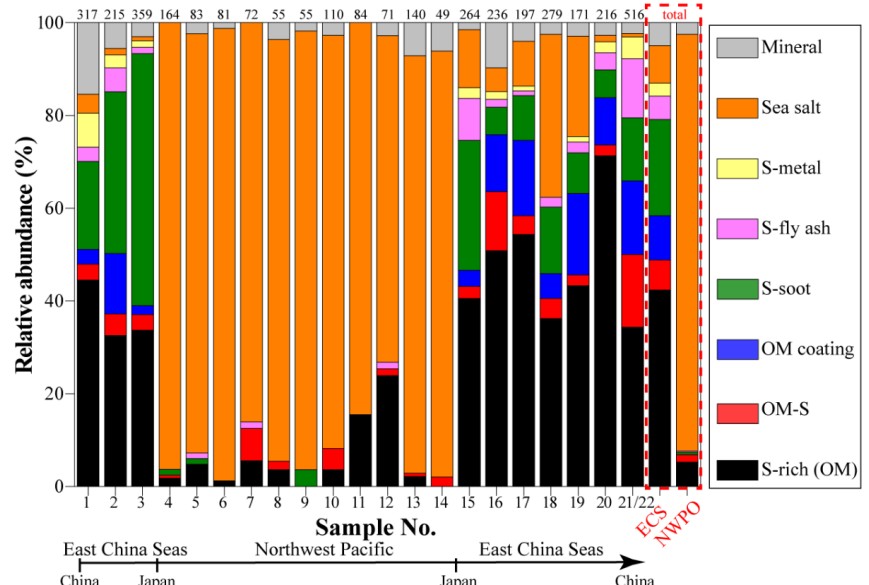


Figure 3. Relative abundances of eight types of aerosol particles in different samples.
The number of analyzed aerosol particles is shown above the column.





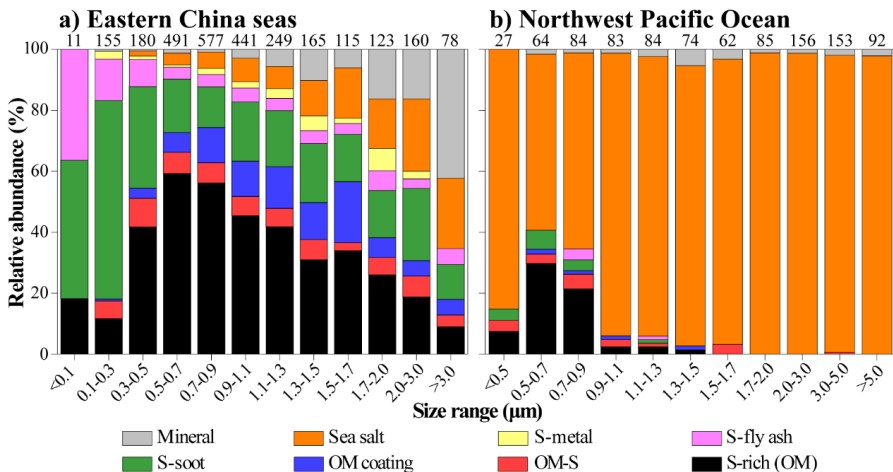

Figure 4. Relative abundances of individual particles in different size bins.

Figure 5. Morphology and EDS spectra of the typical fresh, partially aged, and fully

aged SSA. The main anionic elements are shown in the square brackets.

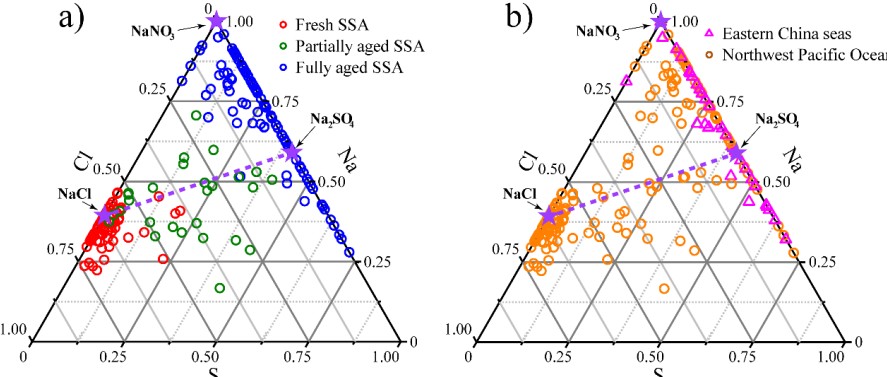


Figure 6. Triangular diagram of Na-Cl-S from EDS data (weight percentage) showing
the elemental composition of SSA particles. The three stars represent pure NaCl,
$Na_2SO_4$, and $NaNO_3$, respectively. The dash line indicates that Na:Cl:S is changed
only by the postulated reaction of $2NaCl + H_2SO_4 \rightarrow Na_2SO_4 + 2HCl(g)$ (Zhang et
al., 2003). Particles above the dash line are those which S cannot compensate Cl
losses and there should be other processes causing Cl-depletion.

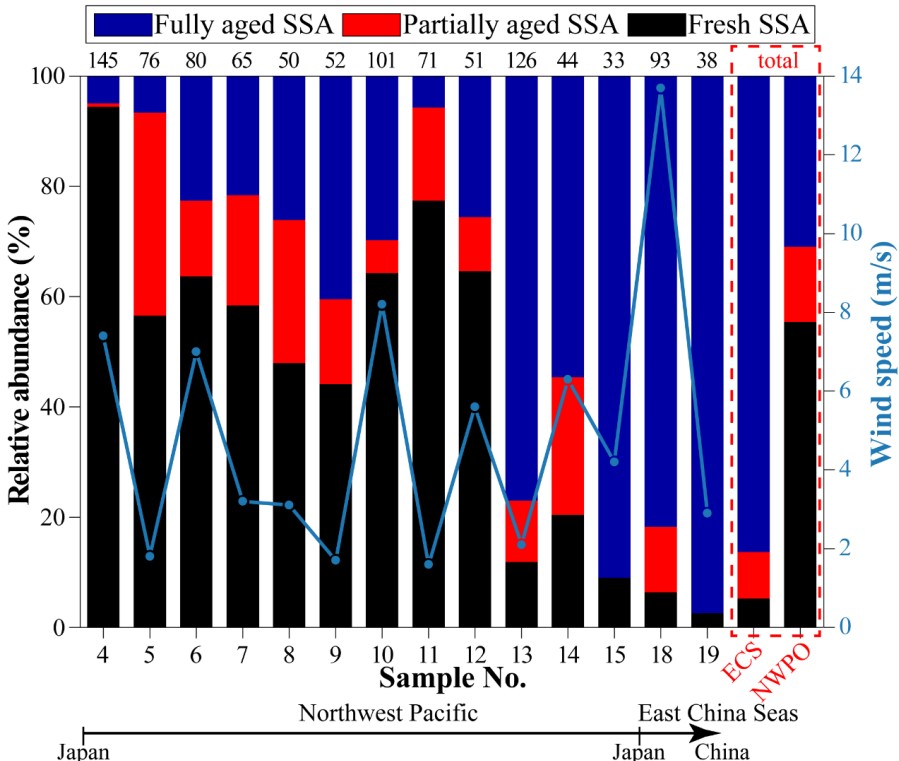

Figure 7. Relative abundances of the fresh, partially aged, and fully aged SSA particles in different samples. Samples with SSA particle less than 30 are excluded due to the small number. The line indicates the wind speed of the corresponding sample.




Figure 8. Relative abundances of three types of SSA particles in different size bins


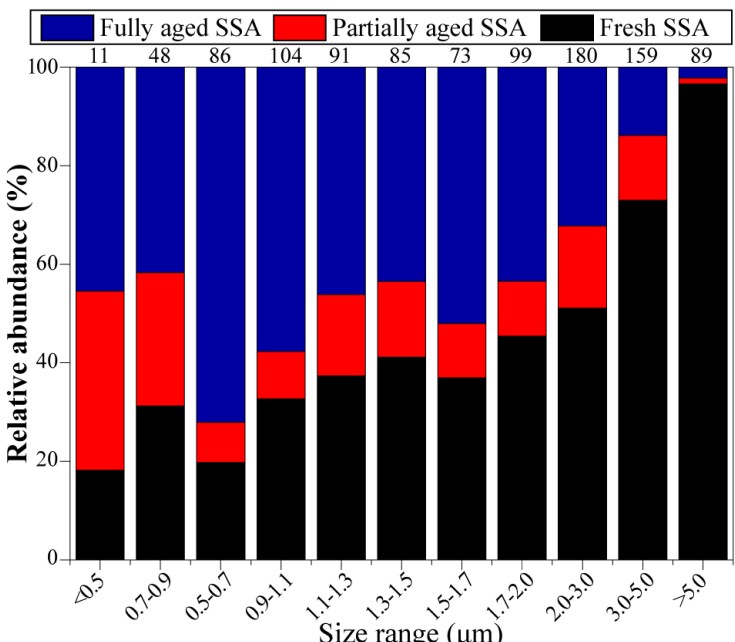


Figure 9. Schematic diagram showing the impact of long-range transported

anthropogenic air pollutants on marine aerosols
