# Peer review of "Long-range transport of anthropogenic air pollutants into the"

_Atmospheric Chemistry and Physics, 2021_

## Author Response (AR1)

**General Response: We thank the Referee for your helpful comments. We have addressed all comments and provided point by point response below. The revised manuscript is presented in below Response.**

This manuscript presented the characteristics of aerosol particles collected from eastern China sea (ECS) and northwestern Pacific Ocean (NWPO) based on particle analysis by TEM. The results showed during the cruise the samples collected in ESC contained more anthropogenic particles than those in NWPO. The chloride depletion in the sea salt particles was also discussed. The manuscript find that the particles may have been impacted by the long range transported anthropogenic pollutants. This study provides a case study in these areas for the better understanding in the impact of anthropogenic emission on the marine environments. The scope of this manuscript is suitable for this journal. A set of issues and comments need to be considered before publication.

Answer: We appreciated the Referee#1's comments which significantly improve the quality of the manuscript. We carefully answer them one by one as below. The modifications were highlighted in red in the revised manuscript.

Comments:

1, The manuscript presented a new set of data for this area, but the general implications in the current version are somewhat limited or not clearly discussed. It is suggested to re-categorize the manuscript as a measurement report. The manuscript stated that the acidic gas precursors transported for a longer distance and thus posed a significant impact on particles in NWPO which can't directly concluded from the evident or data of particle characteristics presented. The conclusion in Line 31-35 is overstated. In addition, in those 11 samples from NWPO, there are only two samples that contain S-rich particles over 15% in number. Also, most of these from 11 samples from NWPO are originated from Sea of Japan and were transported to NWPO within two days, that means those sea salt particles may have been aged when passing Japan.

Answer: (1) Our primary thought about this paper was to discuss the impacts of long-range transported anthropogenic air pollutants to the marine air. We carefully revised the paper and significantly improved the quality. We believe that the revised manuscript is more suitable as a research article.

(2) The conclusion in Line 31-35 was revised to avoid overstatement.

Context: Our results highlight that anthropogenic acidic gases in the troposphere (e.g., $SO_2$, $NO_x$, and volatile organic compounds) could be transported to remote marine air and exert a significant impact on aging of SSA particles in the NWPO. The study shows that anthropogenic particles and gases from East Asia significantly perturb different aerosol chemistry from coastal to remote marine air.

(3) Samples 11 and 12 contain S-rich particles over 15% in number. The high S-rich fraction in these two samples was attributed to new particle formation event caused by long-range transported continental gases (Zhu et al., 2019, New particle formation in the marine atmosphere during seven cruise campaigns, Atmospheric Chemistry and Physics).

(4) If sea salt particles were from Sea of Japan, passing through Japan to our sampling site, we would expect more anthropogenic aerosols (e.g., S-rich, soot, metal, etc.) found in the NWPO samples. However, anthropogenic aerosols accounted for small fraction.

We do think that sea salt particles aging when passing Japan was not the case in our study.

2, Line 27-28, I didn't find the definition for the primary and secondary anthropogenic aerosols in the text based on these TEM analysis or particle classification.
Answer: In this study, primary anthropogenic aerosols include OM, soot, metal, and fly ash particles. Secondary anthropogenic aerosols are S-rich and OM coating particles. In the manuscript, we revised the "primary and secondary anthropogenic aerosols" to "anthropogenic aerosols" and provided a brief definition of anthropogenic aerosols in Abstract and Section 3.1.
Context: Anthropogenic aerosols (e.g., sulfur (S)-rich, S-soot, S-metal/fly ash, organic matter (OM)-S, and OM coating particles) were identified and dramatically declined from 87% to 8% by number from the ECS to remote NWPO.

3, Line 35-38, The statement doesn't provide useful information or conclusion for this manuscript.
Answer: We revised the sentence.
Context: More attention should be given to the modification of SSA particles in remote marine areas due to the influence of anthropogenic gaseous pollutants.

4, Line 49-51, Chloride depletion doesn't release I-containing compounds.
Answer: Sorry for the misleading, we removed the sentence from the manuscript.

5, Line 55-58, The references are not related to ice nuclei and so the statement need more related references for aged SSA serving as IN.
Answer: We added more references about SSA serving as IN.
    Kanji, Z. A., L. A. Ladino, H. Wex, Y. Boose, M. Burkert-Kohn, D. J. Cziczo, and M. Krämer (2017), Overview of Ice Nucleating Particles, Meteorological Monographs, 58, 1.1-1.33.
    Kong, X., M. J. Wolf, M. Roesch, E. S. Thomson, T. Bartels-Rausch, P. A. Alpert, M. Ammann, N. L. Prisle, and D. J. Cziczo (2018), A continuous flow diffusion chamber study of sea salt particles acting as cloud nuclei: deliquescence and ice nucleation, Tellus B: Chemical and Physical Meteorology, 70(1), 1-11.

6, Line 72, Please rewrite this sentence, it is not very clear what that means.
Answer: We revised the sentence for better understanding.
Context: Therefore, it is important to understand the physicochemical properties of long-range transported anthropogenic aerosol particles in marine air.

7, Line 152-155, Line 176-180, the classification scheme is not clear and somehow mixed. Please use the same definition for the particle types. Line 157-159, how does two types of OM particles were distinguished based on the TEM analysis?
Answer: Sorry for the confusion.
S-rich, OM, soot, metal, fly ash, mineral , and sea salt (Line 152-155) are seven basic aerosol components in this study. However, most of the aerosol particles observed in this study contained two or more different aerosol components. We further defined six internally mixed particles to elucidate the mixing states of aerosol particles (Line 176-180). S-metal refers to metal particles mixed with sulfate (Figure 2c). S-fly ash refers to fly ash particles mixed with sulfate (Figure 2d). S-soot refers to soot particles mixed with sulfate (Figure 2e). OM coating refers to secondary organic matter coated on

sulfate (Figure 2f). OM-S refers to primary OM particle mixed with sulfate. S-rich refers to secondary inorganic particles (e.g., $(NH_4)_2SO_4$ and $NH_4NO_3$), which are formed from their gaseous precursors, such as $SO_2$, $NO_x$, and $NH_3$ (Figure 2h).

In the TEM images, the POM particles normally have a spherical or irregular shape. The SOM particles display a core-shell structure, which usually represents an inorganic core (e.g., sulfate) coated by secondary organics.

We also revised these sentences in the manuscript for clear understanding.
Context: To elucidate the mixing structure of the non-SSA particles, we further defined six types of internally mixed particles: S-metal, metal particles mixed with sulfate (Figure 2c); S-fly ash, fly ash particles mixed with sulfate (Figure 2d); S-soot, soot particles mixed with sulfate (Figure 2e); OM coating, secondary organic matter coated on sulfate (Figure 2f); OM-S, primary OM particle mixed with sulfate (Figure 2g); and S-rich, secondary inorganic sulfate and nitrate particles (e.g., $(NH_4)_2SO_4$ and $NH_4NO_3$) (Figure 2h).

In the TEM images, the POM particles normally have a spherical or irregular shape. The SOM particles display a core-shell structure, which usually represents an inorganic core (e.g., sulfate) coated by secondary organics (Li et al., 2016b).

8, Line 198-199, Is the 5% significantly higher than 2% based on the number of particles analysis using TEM? Have you considered the uncertainties in the classification since the total number of particles analyzed are low?
Answer: Although the total number of mineral particles analyzed in this study is small, the uncertainty in mineral particles based on manual identification is low. We can clearly distinguish mineral particles under high-resolution TEM due to their unique morphology and chemical composition. In order to avoid potential misunderstanding, we revised the "twice as high as" to "higher than".
Context: Altogether, the number fraction of mineral particles was 5% in the ECS, higher than that in the NWPO.

9, Line 231, there is no evidence showing that S-rich particles are formed from the precursors.
Answer: Sorry for the confusion. We acknowledge that we do not have evidence to prove it. Thus, we removed the sentence from the manuscript.

10, Line 310, This statement doesn't need a figure here. There is no further description for this Figure 9.
Answer: We added one more paragraph to describe the schematic diagram in Figure 9.
Context: Based on the results and discussion above, a conceptual model was proposed to summarize the impact of long-range transported anthropogenic air pollutants on marine aerosols. Both anthropogenic gases and aerosol particles could be transported to downwind marine areas. Anthropogenic aerosol particles from the continent significantly influence the ECS air. During the transport, aerosol particles could be scavenged due to dry or wet deposition while some reactive gases can be transported further to the NWPO air and influence the aging of SSA particles.

11, Line 323-326, It is not clear what does the authors try to explain.
Answer: Sorry for the confusion.

Particles above the dash line represent that Cl in the SSA particles was not only replaced by S deposition. For the fully aged SSA particles, 70% of them in the ECS were above the dash line, while the proportion increased to 87% in the NWPO. The result indicates that S deposition could not compensate Cl-depletion in most of the fully aged SSA particles in the ECS and NWPO. There must be other acids leading to Cl-depletion in the fully aged SSA particles besides reaction with $H_2SO_4$.

Context: Thus, particles above the dash line represent that Cl in these SSA particles was not only replaced by S deposition, other chemical processes (e.g., react with $HNO_3$ and organic acids) also contributed to the Cl-depletion. The number of fully aged SSA particles above the dash line was further counted. For the fully aged SSA particles, 70% of them in the ECS were above the dash line, while the proportion increased to 87% in the NWPO (Figure 6). The result indicates that S deposition could not compensate for Cl-depletion in most of the fully aged SSA particles in both ECS and NWPO. There must be other acids leading to Cl-depletion in the fully aged SSA particles besides the reaction with $H_2SO_4$.

Here we also revised the Figure 6.

[Figure]

Figure 6.

**Anonymous Referee #2**

**General Response: We thank the Referee for your helpful comments. We have addressed all comments and provided point by point response below. The revised manuscript is presented in below Response.**

This study investigates long-range transport of particles into the marine air and aging of sea salt particles. The authors investigated samples from research ship from the Eastern China seas (ECS) to the Northwestern Pacific Ocean (NWPO). They utilized transmission electron microscopy (TEM) to study the morphology and elemental composition of individual particles. They classified particles into different groups and found that S-rich organic matter and S-rich soot (anthropogenic aerosol, ~87%) dominated in the ECS, while sea salt particles were most abundant (~90%) in the NWPO. They further investigated aging of sea salt particles and classified them as fresh sea salt, partially aged and fully aged salt particles. They found higher number fraction of fully aged sea salt particles in the ECS site (86%) compared to the NWPO site (31%). The authors argue that significant Cl depletion of particles at the NWPO site and presence of minor fraction of anthropogenic aerosol, indicated anthropogenic gases transported to the site and responsible for the ageing of sea salt particles.

While the study is interesting, but several issues need to be addressed before it can be considered for publication.

Answer: We appreciated the Referee#2's comments which significantly improve the quality of the manuscript. We carefully answer them one by one as below. The modifications were highlighted in red in the revised manuscript.

Specific comments:

1. One of the main highlights of this study is that anthropogenic gases transported longer distances compared to the anthropogenic aerosol. However, the results are not supported by any gas phase measurements. The authors should be carefully about some of the bold claims.

Answer: Indeed we do not have in-situ gas concentration data to discuss its transport. In order to solve the problem, we acquired satellite observation data to supplement our results (Figure S1).

[Figure]

Figure S1. Time averaged map of $SO_2$ column mass density and $NO_2$ tropospheric column during our observation (17 March to 22 April 2014). The images were

downloaded from NASA Giovanni website (http://giovanni.gsfc.nasa.gov/giovanni) (Acker and Leptoukh, 2007).

2. The authors talk about primary and secondary aerosol, but their differentiation is not supported by any quantitative number. The authors should discuss about the O:C ratio. Also, particle classification sounds more subjective. The classification of different aging of salt particles is based on morphology and elemental composition. Perhaps a fixed criterion of elemental ratios would be useful to classify them along with morphologies.

Answer: I really appreciated the referee#2's comments. In my opinion, different instruments might have their own advantages to analyze aerosol particles. TEM/EDS is not good at quantifying the O and C because the carbonaceous substrate might affect its peak in the elemental spectra. In the individual particle analyses, integration of particle morphology and elemental composition can be an effective way to classify the particle type. We also want to make clear here about the data process. For the TEM analyses, we all manually classified the particles into different particle types based on their morphology and major compositions. The work is different from CCSEM and online instruments that can obtain compositions of large number of particles. Then they can process the semi-quantitative or quantitative data to classify particles even they could not concern the particle morphology. Therefore, I can say that these are different methodologies to identify particle types. It is hard to find a fixed ratio of elements to classify SSA aging degrees. For example, the elements in the center and periphery of a partially aged SSA are different (see Figure 5b). Here the best way is to combine elements and morphologies to identify SSA particles.

    Although the classification of SSA particles was manual work, we distinguish the SSA aging degree following the rules in Section 3.4 to minimize the bias. The method was also applied in our previous paper (Chi et al., 2015, Sea salt aerosols as a reactive surface for inorganic and organic acidic gases in the Arctic troposphere, Atmospheric Chemistry and Physics).

3. The discussion of size dependent aging is not convincing. Some discussion is needed regarding possible Cl depletion of those particles by organics or sulfate? In general, do you see more organic coating for smaller particles?

Answer: Thanks. We try to improve the discussion part. In this study, we concerned the Cl-depletion of sea salts particles based on their elemental composition and morphology (Figure 4), although we did not analyze the organic species in aged particles. Here TEM images could not directly display possible organic coating for smaller SSA particles. But from the SSA aging reaction with organics, we could expect the possible organic coating. In addition, the previous literature used NanoSIMS to directly obtain the organic species mixed in aged sea salts (Chi et al., ACP, 2015). The literature also indicated that the organics should be prevalent in the aged particles.

    If we clearly identify particle types (e.g., fresh, partially aged, and fully aged sea salt), we can obtain particle aging properties among different sizes. For individual

particle analysis, particle number fractions among different size bins can be used to reflect properties of particle aging, as shown in Figure 8.

[Figure]

Figure 8.

4. The authors talk about long-range transport, but discussion of air mass transport and ageing time are vague.
Answer: Based on the HYSPLIT backward trajectories, the air masses in the ECS samples were mainly from eastern China. Most of air masses in the NWPO samples originated from northwest, passing through Japan to sampling location. We added more description to discuss air mass transport.
The aging time is hard to quantify due to the limitation of HYSPLIT. We roughly counted the transport time of the ECS samples is 24 to 48 hours.
Context: Indeed, the HYSPLIT backward trajectories show that the air masses in the ECS samples were mainly from eastern China (Figure 1).
Most of air masses in the NWPO samples originated from northwest, passing through Japan to sampling location within 48 hours.

5. What about the background ozone concentration at the sites? Aqueous sea salt particles, in presence of light and ozone can undergo rapid Cl2 formation via OH generation in the aqueous phase photochemistry of dissolved ozone.
Answer: Indeed, Cl$_2$ formation due to ozone in aqueous SSA particles under UV could lead to Cl-depletion. Unfortunately, we do not have in-situ background ozone concentration data to discuss its effect on SSA aging.

Here we obtain one ozone mapping in East Asia, the ozone concentration during our cruise did not show huge difference from satellite observation (see figure below). Therefore, we can conclude that ozone might be one of the factor causing Cl-depletion, but it should not be the major reason leading to huge difference in SSA aging degrees in different NWPO samples.
The image was downloaded from NASA Giovanni website,
http://giovanni.gsfc.nasa.gov/giovanni
We added one sentence to make explanation about the ozone in the manuscript.

[Figure]

Figure S1c

6. The meteorological data is not discussed properly. Some of samples were collected at high RH conditions that may have aqueous sea salt particles, how that affect the aging of particles and transport of gases.
Answer: Thanks. This is an interesting point which we also concerned in this work. It is well known that the deliquescence relative humidity (DRH) of NaCl is near 75%, but its efflorescence relative humidity (ERH) is 44%. In our study, the sampling RH were higher than 40%, more than half of them were collected at RH above 60%. Thus, in the humid marine air, SSA particles could exist as aqueous droplets, at least their surfaces may be liquid-like due to the existence of hydrates (e.g., $MgCl_2$).
We added one paragraph to discuss comments in Question 5 and 6.
Context: Meteorological conditions also play an important role in SSA particles aging. The hygroscopic cycle of the pure SSA particles shows the deliquescence relative humidity (DRH) near 75% and its efflorescence relative humidity (ERH) near 44%. However, the natural SSA particles begin to take up water at 57% and form a liquid layer on particles due to various inorganic compounds (e.g., $MgSO_4$, $MgCl_2$, and $CaCl_2$) (Wise et al., 2009). In our study, the sampling RH values were higher than 40% and more than half of them were collected at RH above 60% (Table S1). Therefore, most SSA particles should exist as aqueous droplets during the particle hygroscopic cycle, or at least particle surfaces kept the aqueous phase due to the existence of various inorganic compounds (Wise et al., 2009). One previous study showed that Cl-depletion in the aqueous SSA particles due to ozone under UV can produce $Cl_2$ (Oum et al., 1998).

However, the averaged ozone concentration ranging from 42 to 46 ppb during our cruise did not exhibit large differences from the satellite observation in marine air (Figure S1c). Therefore, we conclude that ozone might be one factor causing Cl-depletion in aqueous SSA particles but should not be the major reason leading to the variations of SSA aging degree in different NWPO samples.

7. Samples from ECS to NWPO is very confusing. Please note the Fig.3 and Fig. 7 should have same notations. It seems like some of the samples that show higher aging of sea salt particles (samples 13, 14 and 15 in Fig 7). Are those same samples that also contain higher number fraction of anthropogenic particles?

Answer: Sorry for the confusion. Sample 15 should be classified into the ECS sample, but we mistakenly labeled it into the NWPO in Figure 7. This is the reason why Figure 3 and Figure 7 have different notations. We have corrected the error in Figure 7.

For Samples 13, 14, and 15, only Sample 15 contain higher fraction of anthropogenic particles. Because Sample 15 is closer to continent than other two samples. This also confirms our perspective that anthropogenic gases could be transported farther than aerosols in marine air.

[Figure]

Corrected Figure 7.

8. The average number fraction of fully aged and partially aged sea salt particles seems ~45-50% from Fig. 7. Are there any bias for some of the high Cl depleted samples collected at the NWPO site?

Answer: In total, the fully aged and partially aged SSA particles accounted for 43% in the NWPO samples. The high resolution TEM images can obtain particle details, so it is no problem to identified different types of aged SSA. Due to the manual operation and labor work, we could not analyze large number of particles in the samples. As the reason, we randomly selected one area from center to periphery of the TEM grid which can fully cover the coarse and fine particles. We generally analyze all the particles in the area. The analyzed procedure can guarantee us to obtain enough particles and well represent all sized particles in the samples. We manually and carefully exam every single particle to minimize the bias.

**Anonymous Referee #3**

General Response: We thank the Referee for your helpful comments. We have addressed all comments and provided point by point response below. The revised manuscript is presented in below Response.

The study reports the observation of aerosol particles collected in the East China Sea (ECS) and the northwestern Pacific Ocean (NWPO) onboard a research vessel. TEM was applied to investigate the physicochemical properties of the aerosol particles over the ocean. The authors reported that more anthropogenic particles were found in the ECS than NWPO, and they further discussed the Cl-depletion in the sea salt particles. This paper provides valuable information for understanding the influence of long-range transported anthropogenic pollutants on the marine environment. The paper addresses relevant scientific scope within the scope of ACP. I would like to suggest it to be accepted for publication after some minor revisions.

Answer: We appreciated the Referee#3's comments which significantly improve the quality of the manuscript. We carefully answer them one by one as below. The modifications were highlighted in red in the revised manuscript.

Comments:

1. Line 72 and Lines 99-100, Please rewrite these sentences for an easy way to understand.

Answer: We revised the sentence in Line 72 for better understanding. The sentence in Line 99-100 was removed from the manuscript.

Context: Therefore, it is important to understand the physicochemical properties of long-range transported anthropogenic aerosol particles in marine air.

2. Lines 126-132, the authors should clarify how they chose the observation areas on the TEM grids for the TEM analysis.

Answer: We added more description about choosing the observation areas in the Section 2.1.

Context: Coarser particles are near the center of the sampling spot and finer particles are on the periphery. Therefore, we selected three or four areas from the center to the edge to guarantee the representativeness of the analyzed particles.

3. Lines 177-180, please explain in a detailed way about the classification of internally mixed particles.

Answer: Most of the aerosol particles observed in this study contained two or more different types of aerosols. Thus, we further defined six internally mixed particles to elucidate the mixing states of aerosol particles. S-metal refers to metal particles mixed with sulfate (Figure 2c). S-fly ash refers to fly ash particles mixed with sulfate (Figure 2d). S-soot refers to soot particles mixed with sulfate (Figure 2e). OM coating refers to secondary organic matter coated on sulfate (Figure 2f). OM-S refers to primary OM

particle mixed with sulfate. S-rich refers to secondary inorganic particles (e.g., (NH4)2SO4 and NH4NO3), which are formed from their gaseous precursors, such as SO2, NOx, and NH3 (Figure 2h).

We also revised the sentence for clear understanding.

Context: To elucidate the mixing structure of the non-SSA particles, we further defined six types of internally mixed particles: S-metal, metal particles mixed with sulfate (Figure 2c); S-fly ash, fly ash particles mixed with sulfate (Figure 2d); S-soot, soot particles mixed with sulfate (Figure 2e); OM coating, secondary organic matter coated on sulfate (Figure 2f); OM-S, primary OM particle mixed with sulfate (Figure 2g); and S-rich, secondary inorganic sulfate and nitrate particles (e.g., $(NH_4)_2SO_4$ and $NH_4NO_3$) (Figure 2h).

4. Line 180, it would be better if the authors briefly define anthropogenic aerosols at first.

Answer: We provide a brief definition of anthropogenic aerosols.

Context: Anthropogenic aerosols are particles originated from human activities, including S-rich, S-soot, S-metal/fly ash, OM-S, and OM coating particles in this study.

5. Line 234, a "." is missing in the "e.g.,".

Answer: Corrected.

6. Figure 8, 0.7-0.9 and 0.5-0.7 in the x-axis are reversed.

Answer: Corrected.

7. Further description on Figure 9 should be provided in the main text.

Answer: We added one more paragraph to describe the schematic diagram in Figure 9.

Context: Based on the results and discussion above, a conceptual model was proposed to summarize the impact of long-range transported anthropogenic air pollutants on marine aerosols. Both anthropogenic gases and aerosol particles could be transported to the downwind marine air. Anthropogenic aerosol particles from the continent significantly influence the ECS air. During the transport, aerosol particles could be scavenged due to dry or wet deposition while some reactive gases can be transported further to the NWPO air and influence the aging of SSA particles.